# *Alternaria alternata* Isolated from Infected Pears (*Pyrus communis*) in Italy Produces Non-Host Toxins and Hydrolytic Enzymes as Infection Mechanisms and Exhibits Competitive Exclusion against *Botrytis cinerea* in Co-Infected Host Fruits

**DOI:** 10.3390/jof9030326

**Published:** 2023-03-07

**Authors:** Stefany Castaldi, Jesús G. Zorrilla, Claudia Petrillo, Maria Teresa Russo, Patrizia Ambrosino, Marco Masi, Alessio Cimmino, Rachele Isticato

**Affiliations:** 1Department of Biology, University of Naples Federico II, Complesso Universitario Monte S. Angelo, Via Cinthia 4, 80126 Naples, Italy; 2Department of Chemical Sciences, University of Naples Federico II, Complesso Universitario Monte S. Angelo, Via Cinthia 4, 80126 Naples, Italy; 3Allelopathy Group, Department of Organic Chemistry, Facultad de Ciencias, Institute of Biomolecules (INBIO), University of Cadiz, C/Avenida República Saharaui, s/n, 11510 Puerto Real, Spain; 4AGRIGES SRL, 82035 San Salvatore Telesino, Italy; 5Interuniversity Center for Studies on Bioinspired Agro-Environmental Technology (BAT Center), 80055 Portici, Italy; 6National Biodiversity Future Center (NBFC), 90133 Palermo, Italy

**Keywords:** *Alternaria alternata*, pear pathogens, enzymatic activities, toxins, necrotic activity, *Botrytis cinerea*, competitive exclusion, multi-infection

## Abstract

*Alternaria alternata* is one of the most devastating phytopathogenic fungi. This microorganism causes black spots in many fruits and vegetables worldwide, generating significant post-harvest losses. In this study, an *A. alternata* strain, isolated from infected pears (*Pyrus communis*) harvested in Italy, was characterized by focusing on its pathogenicity mechanisms and competitive exclusion in the presence of another pathogen, *Botrytis cinerea*. In in vitro assays, the fungus produces strong enzymatic activities such as amylase, xylanase, and cellulase, potentially involved during the infection. Moreover, it secretes four different toxins purified and identified as altertoxin I, alteichin, alternariol, and alternariol 4-methyl ether. Only alteichin generated necrotic lesions on host-variety pears, while all the compounds showed moderate to slight necrotic activity on non-host pears and other non-host fruit (lemon, *Citrus limon*), indicating they are non-host toxins. Interestingly, *A. alternata* has shown competitive exclusion to the competitor fungus *Botrytis cinerea* when co-inoculated in host and non-host pear fruits, inhibiting its growth by 70 and 65%, respectively, a result not observed in a preliminary characterization in a dual culture assay. Alteichin and alternariol 4-methyl ether tested against *B. cinerea* had the best inhibition activity, suggesting that the synergism of these toxins and enzymatic activities of *A. alternata* are probably involved in the competitive exclusion dynamics in host and non-host pear fruits.

## 1. Introduction

*Alternaria alternata* is a filamentous ascomycete, with a widespread distribution in nature, occurring on plants as a pathogen and endophyte and in the soil as a saprophyte [1]. The fungus has also been associated with human infections, especially in immunocompromised patients, and even occasionally in healthy hosts [2].

As a phytopathogen, this fungus infects diverse plant species of agronomical interest, such as potatoes, apples, pears, tomatoes, strawberries, and blueberries. Causing necrotic lesions, *A. alternata* infections lead to food inedibility and crop losses [3]. The first disease of European pear (*Pyrus communis* L. var. sativa de Candolle) associated with *A. alternata* was found on cv. Le Lectier in orchards in Japan, in 1993. Then, black spot lesions on pear leaves, typical of this pathogen, were found in Italy and Korea [4]. *Alternaria* spp. produce a broad arsenal of metabolites that exhibit a variety of biological activities, such as phytotoxic, cytotoxic, and antimicrobial properties [5]. Like many phytopathogenic fungi, *Alternaria* spp. can produce host-selective toxins (HSTs), an ability that has been strictly associated with fungal pathogenesis [6,7]. Moreover, HSTs are essential determinants of host range and specificity in particular plant species and cultivars. For example, some *A. alternata* pathotypes produce AK-toxin causing a black spot on Japanese pear fruit, while others, the AAL and AF toxins, infect tomatoes and strawberries, respectively [8]. All the HSTs have different modes of action, causing biochemical and genetic modifications in the host [9]. The AK-toxin, AF-toxin, and AC-toxin can lead to DNA breakage and apoptotic cell death, interrupting plant physiology by mitochondrial oxidative phosphorylation and affecting membrane permeability, with a devastating effect on plants.

Besides HSTs, *A. alternata* produces non-specific host toxins (NSTs) and cell-wall-degrading enzymes (CWDEs), both required for its full virulence. NSTs are thought to contribute to some virulence features, such as symptom development and in planta pathogen propagation [7], and CWDEs are probably involved in the first step of plant attacks (host penetration), but the exact pathological roles of NSTs and CWDEs have yet to be well characterized.

This lack of information is due to the primary role of HSTs in fungal pathogenesis, which often masks the functions of NSTs and CWDEs. Therefore, the toxic effects of *A. alternata* metabolites due to NSTs and CWDEs have received minor attention compared to those reported for HSTs mycotoxins [10]. A wider study of the activity of isolated fungal metabolites can allow the identification of compounds directly related to the pathogenic activity of the fungus, making it possible to create chemo-libraries that facilitate the linking of the structure of the compounds with the species that produce them, and their effect on host and non-host crops, as well as their biosynthetic features [11]. In this context, we focused on NSTs and CWDEs used by the *A. alternata* RS strain isolated from infected pears in Italy, and characterization of hydrolytic enzyme activities of *A. alternata* RS and the identification of the metabolites produced in vitro were undertaken. The phytotoxic activity of the isolated compounds was evaluated on pear (host and non-host varieties) and lemon fruits [12].

Since, in nature, plants are infected by multiple pathogen species/genotypes [13,14], we also investigated the interaction between the strains of *A. alternata* RS and *Botrytis cinerea*, both pathogenic agents attacking *Pyrus communis*. *B. cinerea* is one of the most destructive fungal pathogens, affecting numerous plant hosts [15]. Recently, in Turkey, *B. cinerea* caused significant amounts of pre-harvest rot in pear orchards [16].

An infection study of one host by multiple pathogens is of particular interest since it allows us to understand the pathogens’ virulence and highlights alteration of the normal course and severity of the disease caused by a single pathogen. Moreover, multiple-infection studies can lead to the identification of novel antimicrobial compounds for more sustainable control of pathogen complexes in agricultural systems [17]. For example, some *Pseudomonas* strains secrete antimicrobial compounds that antagonistically affect co-pathogens [18]. Some compounds are not phytotoxic and have been proposed as bio-based anti-microbials for sustainable plant pathogen control [19,20,21].

The two pathogen strains examined in our work revealed antagonistic traits depending on the performed assays. Interestingly, one of the identified NST toxins had an inhibitory effect on *B. cinerea* and did not cause damage to the host and non-host plants, suggesting its potential use as a bio-pesticide.

## 2. Materials and Methods

### 2.1. Fungal Isolation

*Alternaria alternata* RS used in this study was originally isolated from fruits of Abate Fetel pear tree (*Pyrus communis*) showing brown spot disease symptoms sampled in Terrazzo (Verona, Italy) in 2019. The strain was stored on Potato Dextrose Agar (PDA) (Difco) plates in the culture collection of Agriges s.r.l., San Salvatore Telesino, Benevento, Italy.

### 2.2. Morphological Characterization

The phenotypic variant of the fungal isolate was determined by visual inspection. The fungus was grown in PDA broth at 28 °C for 7 days, then the Petri plate was photographed. The hyphae and conidia from the fungal cultures were stained with lactophenol blue solution (LPCB), Sigma-Aldrich, Saint Louis, MO, USA) and observed under a phase-contrast light microscope using an Olympus BX51 with a 60× objective UPlanF1.

### 2.3. DNA Extraction and Identification

DNA extraction from the fungal mycelium was performed as described by Stirling D. (2003) [22]. Briefly, the fungus was grown in 150 mL of potato dextrose broth (Difco) for 7 days at 28 °C and shaken at 150 rpm. The mycelium was harvested by filtration, and 0.2 g of dry weight was ground in a mortar and resuspended in 5 mL of CTAB extraction buffer. After incubation at 65 °C in a water bath for 30 min, an equal volume of chloroform/isoamyl alcohol (24:1) was added and centrifuged at 2000× *g* for 10 min at room temperature. Later, an equal volume of isopropanol was added to the supernatant, and the precipitated DNA was rinsed with 70% ethanol. After air drying, the DNA was resuspended in sterile water overnight at 4 °C.

For the identification, the ITS region was sequenced using the forward primer ITS1 (50-TCCGTAGGTGAACCTGCGG-30) and the reverse primer ITS4 (50-TCCTCCGCTTATTGATATGC-30) [23]. The PCR conditions were as follows: 94 °C for 10 min, followed by 30 cycles, 92 °C for 1 min, 58 °C for 1 min, 72 °C for 1 min, and final synthesis 72 °C for 5 min. The PCR products were separated by agarose gel electrophoresis. The sequence was performed by Bio-Fab Research (Rome, Italy), and the resulting ITS sequences were analyzed for homologies to sequences deposited in the GenBank (https://blast.ncbi.nlm.nih.gov, accessed on 30 October 2022). The sequence of the strain was deposited in GenBank as *Alternaria alternata* strain RS (Accession number: OP715870).

### 2.4. Enzymatic Activities

*A. alternata* was tested for extracellular enzyme activities in the chromogenic media described by Kwon et al. (2007) [24] with some modifications. Preculturing of *A. alternata* was done on PDA at 28 °C for 7 days. To detect fungal extracellular enzyme activity, the precultures were transferred onto a medium containing Skimmed Milk Agar (SMA) for the protease activity [25], starch from potato (Sigma Aldrich, St. Louis, MO, USA) for amylase activity [26], 0.5% xylan (Megazyme) for xylanase activity [27], and 0.5% carboxymethylcellulose (CMC) (Sigma-Aldrich, USA) for cellulase activity [28,29]. Congo red dye or iodine solution was used for the chromogenic reaction to detect the enzymatic activities. After 5 days of incubation at 28 °C, a halo around the fungal plug was observed and measured, indicative of the hydrolytic activities. Each test was conducted in triplicate and repeated three times. The halo measurement was expressed as the mean of the obtained values.

### 2.5. Dual Culture Assay

To detect the antifungal activities between pathogen fungi that infect the same host plant (Table 1), a dual culture assay was performed as described by Yasmin et al. (2019) [30] with some modifications. A fungal disc (4 mm) was placed at the margin of the PDA plate and 1 cm of the distance from the antagonist fungal disc (4 mm). The plates were incubated at 28 °C for 7 days. Plates inoculated with the fungus alone were used as negative controls. The assay was replicated three times. The percentage growth inhibition was calculated using the following formula:% inhibition = [1 − (Growth of fungus/control growth)] × 100

### 2.6. Pathogenicity Tests on Host and Non-Host Pear Fruits

The pathogenicity of the isolate (Table 1) was also tested on pear fruits from two *Pyrus communis* cultivars (Abate Fetel as a global cultivar and Decana del Comizio as an Italian cultivar). The experiment was performed as described by Aung et al. (2020) [31] with some modifications. Briefly, the fruits were surface-sterilized by dipping in 70% of EtOH for 2 min and then washed with sterilized distilled water 3 times. The fruits were wounded by a puncher (4 mm), and *A. alternata* and *B. cinerea*’s mycelia plugs (4 mm) were cut from the edge of 3-day-old colonies and placed on wounded sites. Sterile PDA plugs were used as controls. To detect the antifungal activity between fungi competitors of pear, the same fruit was wounded with two punchers at 1.5 cm of distance, and the fungal plugs were placed. Each treatment was conducted with three replications and repeated three times (nine pear for treatment). The disease development was registered after 4 days the severity, defined by decay lesion diameter (cm).

The experimental design follows used for both pears cultivars was:

Pear 1: Inoculation of *A. alternata* alone and PDA broth used as a negative control.

Pear 2: Inoculation of *B. cinerea* alone and PDA broth used as a negative control.

Pear 3: Inoculation of both fungi *A. alternata* vs *B. cinerea.*

### 2.7. Chemical Procedures

^1^H NMR spectra were recorded in CDCl_3_, CD_3_OD or (CD_3_)_2_CO, also used as internal standards, at 400 MHz on a Bruker (Karlshrue, Germany) spectrometer. ESI mass spectra were recorded using the LC/MS TOF system Agilent 6230B (Agilent Technologies, Milan, Italy), HPLC 1260 Infinity in positive mode. Preparative and analytical TLC was performed on silica gel (Merck, Kieselgel 60 F_254_, 0.50 and 0.25 mm, respectively) plates (Merck, Darmstadt, Germany), while column chromatography was performed on silica gel (Merck, Kieselgel 60, 0.063–0.200 364 mm). The spots were visualized by exposure to UV light and/or by spraying first with 10% H_2_SO_4_ in MeOH, and then with 5% phosphomolybdic acid in EtOH, followed by heating at 110 °C for 10 min. Sigma-Aldrich Co. (Milan, Italy) supplied all the reagents and solvents.

### 2.8. In Vitro Fungal Growth, Isolation and Chemical Characterization of Fungal Metabolites

The fungus was grown in 2 L of Potato Dextrose Broth (Difco, Tucker, GA, USA) for 15 days at 28 °C with shaking at 150 rpm. The supernatant was obtained by centrifugation at 7000× *g* for 30 min and filtration using 0.22 μm pore diameter membranes (Whatman, Maidstone, UK).

The culture filtrate was collected (pH 7.5), concentrated under reduced pressure, and extracted with EtOAc (3 × 200 mL). The combined organic extracts were dried with Na_2_SO_4_ and evaporated under reduced pressure. The recovered residue (254.0 mg) was fractionated by column chromatography on silica gel eluted with a gradient of CHCl_3_/*i*-PrOH (95:5, 9:1, 7:3, *v*/*v*), yielding eight groups (F1–F8) of homogeneous fractions (F1–F8). The residue of F2 (17.0 mg) was further purified by TLC eluted with CHCl_3_/*i*-PrOH (95:5, *v*/*v*), giving a pure amorphous solid identified as alternariol 4-methyl ether (**4**, 6.6 mg). The residue of F5 (26.1 mg) was further purified by TLC eluted with *n*-hexane/EtOAc (4:6, *v*/*v*) yielding a pure amorphous solid identified as alternariol (**3**, 2.2 mg). The residue of F7 (15.4 mg) was further fractioned by CHCl_3_/*i*-PrOH (95:5, *v*/*v*), providing four fractions (F7A-F7D). The residues of F7A (2.7 mg) and F7B (3.0 mg) were further purified by analytical TLC with CHCl_3_/*i*-PrOH (99:1, *v*/*v*) yielding two pure amorphous solids identified as altertoxin I (**1**, 1.1 mg) and alteichin (**2**, 1.2 mg). The same metabolites (toxins **1**–**4**) were obtained from the organic extracts of other two independent *A. alternaria* growths in 1 L of PD broth. In particular, using the method described in the paragraph. 2.7, we obtained the following compounds:

Altertoxin I, also known as dihydroalterperylenol (**1**): ^1^H NMR spectrum (Appendix A) agreeing with data previously reported [32]; ESIMS (+) *m/z*: 353 [M + H]^+^ (Appendix A), consistent with the molecular formula C_20_H_16_O_6_. [α]^25^_D_ + 378° (c 0.4, CHCl_3_); [α]^25^_D_ + 396° (c 0.39, CHCl_3_) lit. [33].

Alteichin, also known as alterperylenol (**2**): ^1^H NMR spectrum (Appendix A) agreeing with data previously reported [34]; ESIMS (+) *m/z*: 351 [M + H]^+^ (Appendix A), consistent with the molecular formula C_20_H_14_O_6_. [α]^25^_D_ + 672° (c 0.25, acetone); [α]^25^_D_ + 699° (c 0.26, acetone) lit. [34].

Alternariol (**3**): ^1^H NMR spectrum (Appendix A) was in agreement with data previously reported [35,36]. ESIMS (+) *m/z*: 259 [M + H]^+^ (Appendix A), consistent with the molecular formula C_14_H_10_O_5_.

Alternariol 4-methyl ether, also known as alternariol-9-*O*-methyl ether (**4**): ^1^H NMR spectrum (Appendix A) was in agreement with data previously reported [36,37]. ESIMS (+) *m/z*: 273 [M + H]^+^ (Appendix A), consistent with the molecular formula C_15_H_12_O_5_.

### 2.9. Phytotoxic Activity of the Fungal Metabolites

The phytotoxic activity of compounds **1**–**4** was assayed on host (Abate Fetel pear) and non-host (Decana del Comizio pear, and lemon, *Citrus limon*) fruits at 5 × 10^−3^ M and 2.5 × 10^−3^ M concentrations. Samples were dissolved in dimethyl sulfoxide (DMSO, final concentration: 4%) and diluted in distilled water to reach the desired concentration. The surface of three fruits of each species was disinfected with sodium hypochlorite (50 mg/mL) and subsequently washed three times with distilled water. Aliquots of 20 μL of the solutions containing compounds **1**–**4** at the two concentrations were released on the fruits. A solution of 4% of DMSO in distilled was used as negative control, while a solution of the known phytotoxin *epi*-pyriculol (tested at the same concentrations) was used as a positive control [38]. Then, only for the lemon, the surface was punctured three times using the needle of a sterile syringe following the procedure previously reported [39]. The injections were performed at room temperature (i.e., 14–24 °C), and the results were checked daily after inoculation. The experiment was repeated in triplicate and the necrotic lesion development was evaluated using a visual 0–3 scale (0 = no necrosis; 1 = slight necrosis; 2 = intermediate necrosis; 3 = severe necrosis). The effects of the toxins on the fruits were observed up to 10 d.

### 2.10. Antifungal Activity of the Fungal Metabolites against Botrytis cinerea

The antifungal activity of compounds **1**–**4** isolated from *A. alternata* was tested against *B. cinerea,* as reported in Zdorovenko et al. (2021) [40], with some modifications. Compounds **1**–**4**, and the positive control, the fungicide pentachloronitrobenzene (Sigma-Aldrich), were dissolved in 4% of DMSO at a final concentration of 5 × 10^−3^ M. The sensitivity of *B. cinerea* to these compounds was evaluated on PDA as inhibition of mycelial radial growth. In brief, mycelial plugs 4 mm diameter were cut from the margin of actively growing 5-day-old colonies and one plug was placed in the center of the Petri dish with the mycelia in contact with the medium. Then, the different compounds were applied separately on the top of each plug. The negative control was obtained by applying 20 μL of 4% of DMSO. The plates were incubated at 28 °C for 5 days. Inhibition of the fungal growth was observed as a decrease of growth compared to the negative control.

### 2.11. Statistical Analysis

The results of pathogenicity activity analysis are expressed as means of independent experiments  ±  standard errors (SE). Analysis of variance was carried out by one-way ANOVA using GraphPad Prism 8 software.

## 3. Results and Discussion

### 3.1. Preliminary Characterization

A pure culture fungal isolate on PDA media was obtained from brown spots of Abate Fetel pears (*Pyrus communis*) collected in Terrazzo (Verona, Italy) in 2019. The isolate was identified by morphological analysis and DNA sequencing. On PDA plates, the mycelium developed air hyphae on grayish-white colonies that turned olivaceous-black in seven days (Figure 1A). Under an optical microscope, the conidia appeared ovoid or ellipsoidal, pale brown with a cylindrical beak. Short beaks and slim long septate we also observed, Figure 1B,C). The mean conidial dimension was 19.2 × 13.4 μm on PDA. The examined morphological parameters, such as conidia dimensions, conidiophore morphology, number of divisions, and beak structure and colony development, were consistent with the *Alternaria alternata* parameter ranges defined by Simmons (1999) [41], so the fungal isolate was suggested as *A. alternata* (Figure 1). To confirm the identity, the ITS region of DNA was amplified, showing that the fungus corresponded to *A. alternata* with a percentage of identity of 99.60%.

The necrotrophic fungal pathogen *A. alternata* can attack many plant species by producing a broad range of host-selective and non-selective compounds. Cell wall-degrading enzymes (CWDEs) and toxins are the most means by which pathogens penetrate and colonize their host. Therefore, CWDEs and non-specific toxins (NSTs) were analyzed to better understand the mechanisms underlying *Alternaria* pathogenesis.

### 3.2. Enzymatic Activity of A. alternata

Phytopathogenic fungi secrete a broad range of hydrolytic enzymes to break down the complex polysaccharides of the plant cell wall to penetrate the host and develop the disease. Most of the characterized CWDEs are glycoside hydrolases, which can hydrolyze glycosides. The hydrolytic activity of the fungus was tested as reported in the Material and Method section. The results show that the fungus has strong amylase and xylanase activities, which are involved in stealth pathogenicity strategies by fungi to avoid host detection and promote necrosis of the plant tissue surrounding the infected plant areas, respectively [42].

Only moderate cellulase activity was detected, which may be due to the lack the induction of the specific genes, as happens during infections. The fungus did not produce protease enzymes under tested conditions (Table 2).

### 3.3. Dual Culture Assay with Phytopathogenic Fungi Competitors of Pear Fruits

In nature, an individual host plant interacts, simultaneously or successively, with more than one strain of the same pathogen species, and with different pathogen species, causing multi-infections. Furthermore, different kinds of interactions may establish among pathogens infecting the same environmental niche, such as competition, cooperation, or coexistence. We investigated the dynamics of multi-pathogen interactions between *A. alternata* RS and a field strain of *B. cinerea* isolated from pear fruit. With this aim, a dual culture assay was performed between the two pears pathogens. The results in Figure 2 show that *B. cinerea* can inhibit *A. alternata* by 44%.

### 3.4. Pathogenicity Activity of A. alternata in Host Pear Fruits

To analyze the pathogenic effect of *A. alternata* on host pear (Abate Fetel) fruits, a 4 mm fungal plug was inoculated in the fruits. The experiment was also performed with *B. cinerea*. Moreover, a multi-infection with a co-inoculate of these two phytopathogenic fungi on pears fruits was simulated. As shown in Figure 3A, the external appearance of lesions is difficult to appreciate due to the absence of a necrotic area on the tissue. By contrast, internal injuries consisted of black or brownish necrosis that extended from 0.5 to 1.5 cm from the inoculation point (Figure 3B). It was also possible to observe the hardening of the tissue at the necrotic patch. When the fungi were inoculated separately, a severe inner lesion was produced by *B. cinerea* compared to *A. alternata* RS, in agreement with the results previously reported by Lutz et al. (2017) [43].

Unexpectedly, when the fungi were co-inoculated, *A. alternata* inhibited the fungal growth of *B. cinerea* with a reduction of the necrotic area by about 70%. This result suggests a different mechanism of competition between the two pathogens on host pears and in vitro.

### 3.5. Production of Fungal Metabolites In Vitro

To evaluate the production of fungal metabolites potentially involved in the pathogenicity mechanisms, *A. alternata* was grown in vitro as described in the Material and Method section. The EtOAc extract of *A. alternata* culture filtrates was purified following the procedure reported in the same area, obtaining four metabolites belonging to two different classes of natural compounds: perylenequinones (compounds **1** and **2**), and dibenzopyrones (compounds **3** and **4**) (Figure 4) [44].

The isolated compounds were altertoxin I (**1**, also known as dihydroalterperylenol), alteichin (**2**, also known as alterperylenol), alternariol (**3**) and alternariol 4-methyl ether (**4**, also known as alternariol-9-*O*-methyl ether). Compound **4** was isolated with the best yield. The ^1^H NMR spectra and molecular ion peaks obtained for compounds **1** and **2** (Appendix A) indicated that both were perylene derivatives, which are among the common structural classes of the isolated *Alternaria* toxins [14]. Compound **1** was identified as altertoxin I and compound **2** as alteichin by comparing their spectroscopic data (^1^H NMR) with those reported in literature [32,34]. Their identity was confirmed by data obtained from their ESIMS spectra, which showed the protonated [M + H]^+^ ions at *m/z* 353 (for compound **1**) and 351 (for compound **2**). Finally, the specific optical rotation values recorded for compounds **1** and **2** allowed us to unequivocally identify their stereochemistry by comparing them with the values previously reported in the literature [33,34]. On the other hand, the spectroscopic (^1^H NMR) and spectrometric (ESIMS) data obtained for compounds **3** and **4** (Appendix A) indicated that these metabolites were closely related and belong to a class of dibenzopyrone derivatives, which are also produced by of *Alternaria* spp. [14,45]. In particular, the ^1^H NMR spectrum of compound **3** agreed with data previously reported for alternariol (**3**) [35,36], while that of compound **4** was in agreement with data previously reported for one of its analogues, namely its 4-methyl ether (**4**) [36,37]. Their identity was also confirmed by the data obtained from their ESIMS spectra which showed protonated [M + H]^+^ ions at *m/z* 259 (for compound **3**) and 273 (for compound **4**).

Altertoxin I (**1**), alternariol (**3**) and alternariol 4-methyl ether (**4**) are among the most common mycotoxins produced by *Alternaria* spp., and can be found as contaminants of foodstuffs [45,46]. Alteichin (**2**) or alternariol (**3**) have been described as reddish pigments [47]. Compounds **1** and **2** are perylenequinones, a family of natural products whose structure is characterized by a pentacyclic conjugated chromophore that provides light-induced biological activity [48]. The biological activities of perylenequinones have been widely studied in the pharmacological field. Eighty-five percent of the already-known perylenequinones have been discovered from fungal sources, though they have also been found in plants and animals [49]. Altertoxin I (**1**) and alteichin (**2**) are produced by different *Alternaria* species, including fungal pathogens, and phytotoxic activities have been reported for both compounds [45,50,51]. Alternariol (**3**) and alternariol 4-methyl ether (**4**) are dibenzopyrone derivatives, tricyclic aromatic compounds commonly produced by *Alternaria* species [45]. These compounds are mycotoxins that are frequent contaminants in foodstuffs, though the toxicity of alternariols for humans and animals is considered low [52,53]. There are several reports in the literature that have focused on pharmacological activities and biosynthetic aspects of compounds **3** and **4**, and some studies have shown activities of agronomic interest for both toxins [52,54,55].

### 3.6. Pathogenicity Activity of A. alternata in Non-Host Pear Fruits

Since studies on the toxins secreted by *A. alternata* detected the presence of unselective toxins, the same experiment performed in paragraph 3.4 was conducted in non-host pear fruits using the Italian cultivar of pear fruits Decana del Comizio. As shown in Figure 5A, the external appearance of lesions is visible as a black area with *A. alternata*, and a brownish area with *B. cinerea*. The external injuries extended from 0.5 to 2 cm from the inoculation point (Figure 5B), and the internal injuries from 0.8 to 2.4 cm (Figure 5C). In this case, *B. cinerea* was more phytotoxic than *A. alternata*. The co-inoculation of both fungi on the pear fruits confirms the previous data observed in Figure 3. *A. alternata* in the presence of *B. cinerea* showed a competitive exclusion during the co-inoculation, inhibiting the growth of *B. cinerea* by about 65%.

### 3.7. Phytotoxic Activity of the Isolated Compounds

The phytotoxic activity of the isolated compounds (**1**–**4**) was tested against two pear species (Abate Fetel, host; and Decana del Comizio, non-host), and lemon (non-host) fruits, together with the known phytotoxin *epi*-pyriculol [38] as a positive control, at 5 × 10^−3^ M and 2.5 × 10^−3^ M. Different activity levels were obtained according to the fruit at 5 × 10^−3^ M, while no activity was observed for compounds **1**, **3** and **4** at 2.5 × 10^−3^ M. Compound **2** caused slight necrosis only on lemon fruits similarly to the positive control when tested at 2.5 × 10^−3^ M. Observations were daily made for 10 days, though no significant changes were observed from the third day. The positive control always showed necrosis, and null effects were always observed for the negative control. The results obtained using a concentration of 5 × 10^−3^ M are shown in Figure 6 and Table 3.

As shown in Figure 6, visible necrotic effects were observed after treatment when tested at 5 × 10^−3^ M. After 24 h of application, the most active compounds caused mild effects, reaching the maximum level of activity on the third day, as shown in Figure 6 and Table 3. On pear, both on Abate Fetel and Decana del Comizio fruits, after 1 day of treatment alteichin (**2**) was the only compound causing necrosis, at a slight level, and increasing this to intermediate necrosis in the case of the non-host pear after 3 days (Figure 7B,D). Altertoxin I (**1**) also showed activity (slight) after 3 days (Figure 7D) but only on the non-host pear variety (Decana del Comizio). Thus, necrotic activity was only generated by the tested perylenequinones (**1** and/or **2**), the dibenzopyrones **3** and **4** being totally inactive. Though no necrotic lesions were observed after treatment with compound **4** on pear fruits, necrosis activity on pear leaves was reported for this compound [55].

On lemon samples, compounds **1**–**4** showed remarkedly higher necrotic activity. This elevated activity agrees with minor changes in the appearance described for the infected area in pear fruits during the early stages of infection with *A. alternata* [56]. After 1 or 2 days, compounds **2** and **3** showed intermediate necrosis, and compounds **1** and **4** at a slight level. After 3 days, the effects of the activity of all the compounds increased, with compounds **2** and **3** showing severe necrotic effects comparable to those of the positive control (Figure 6F). From the structural point of view, on lemon samples elevated activity between perylenequinones and dibenzopyrones was not observed. It must be highlighted that alteichin (**2**) showed the highest necrotic effects, as also found on pear samples. Furthermore, it must be noted that alteichin (**2**) was the only active compound when tested at the lowest concentration (2.5 × 10^−3^ M) on lemon fruits. Considering the chemical differences between compounds **1** and **2**, the results obtained in our study suggest the importance of the α,β-unsaturated ketone group present in compound **2**. In fact, the absence of this moiety in compound **1** causes a reduction of the phytotoxic activity. It is interesting to note that compound **2** caused necrotic lesions in a previous report on tomato, sunflower, Canada thistle, wheat, and barley leaves [57], and moderate necrotic areas were also found on corn and soybean [50]. The other tested perylenequinone (altertoxin I, **1**), as in our study, also showed selectivity in a previous study that showed remarked necrotic lesions on corn, though no significant lesions for other species such as soybean, crabgrass and timothy [50]. Phytotoxic effects such as necrosis could be directly related with the pro-oxidant properties of the compound [58], so the pro-oxidant activity previously suggested for compound **1** [59] should be noted. Perylenequinones **1** and **2** also exerted phytotoxic activity against seedling growth of lettuce and amaranth (*Amaranthus retroflexus* L.) [51].

Regarding the dibenzopyrone derivatives alternariol (**3**), and alternariol 4-methyl ether (**4**), the higher necrotic effects showed by compound **3** could be explained by the hydroxyl group placed in the position where compound **4** has a methoxy group, hinting that polar groups could have a positive effect on necrosis-inducing activity due to the capacity for forming H-bonds [60]. For both compounds, found in our study as necrotic-causing toxins on lemon fruits but inactive on pear fruits (host and non-host species), previous studies reported activities of interest. Pro-oxidative activity was described for compound **3** [61,62], and it had inhibitory activity against the root growth of *Pennisetum alopecuroides* and *A. retroflexus* when tested at 1000 μg/mL [54]. For compound **4**, necrosis on pear leaves, chlorosis on tobacco leaves, and inhibition of the electron transport chain in isolated spinach chloroplasts would support the toxic potential of this toxin produced by *A. alternata* [52,55].

### 3.8. Antagonistic Effect of Compounds ***1***–***4*** against B. cinerea

The unselective toxins (**1**–**4**) isolated from *A. alternata* were tested for their antifungal activities to investigate their involvement in inhibiting the growth of *B. cinerea* observed in the previous experiments. As a positive control, the commercial fungicide pentachloronitrobenzene (PCNB) was employed. The concentrations tested were those used in the experiment represented in Figure 7.

As shown in Figure 7A,B, when the fungus was treated with different toxins, the growth was slowed. In particular, compounds **2** and **4** decreased the radial growth of the fungus. The mycelium assumed a circular form, and the hyphae showed slow outward growth. Since the radiating hyphae make it easy to move nutrients quickly around the growing mycelium, this could explain the difference observed in plates treated with toxins compared to the control (untreated). The commercial fungicide PCNB completely inhibited the growth of *B. cinerea.*

## 4. Conclusions

The morphological observation of mycelium and the amplification of a fragment of rDNA (including ITS1 and ITS2 and the 5.8S rDNA gene) of a phytopathogenic fungus isolated from infected pear fruits in Italy allowed us to identify the species as *Alternaria alternata*. The fungus showed strong amylase, xylanase and cellulase activities. Moreover, its PDB liquid culture led to the isolation of the four compounds, which were identified as altertoxin I (**1**), alteichin (**2**), alternariol (**3**) and alternariol 4-methyl ether (**4**). When tested on pears (host and non-host varieties) and lemon fruits, the compounds showed different levels of activity. In particular, only compound **2** generated visible necrotic effects on host pears, while on the non-host pear variety, the perylenequinones (**1** and **2**) showed moderate to slight necrotic activity. All the compounds (**1**–**4**) caused severe or moderate necrotic lesions on lemon fruits. These results show that compounds **1**–**4** are non-host-selective toxins and that compound **2** is the most active one.

Furthermore, to understand the strategies used by *A. alternata* in infection dynamics, a multi-infection with the competitor phytopathogen *B. cinerea* was simulated. In our conditions, the co-infection of *A. alternata* with *B. cinerea* depended on the performed assays. When simulated in an in vitro dual culture assay, *B. cinerea* inhibited *A. alternata* growth. In contrast, when replicated in ex vivo experiment (host and non-host pear fruits), *A. alternata* showed a competitive exclusion that inhibited the fungal growth of *B. cinerea* by 70 and 65%, respectively. These preliminary experiments demonstrate that the interaction between the two phytopathogens sharing same ecological niches results in competition, and the most virulent fungus has an intra-host competitive advantage that leads to the exclusion of less virulent fungi. Probably the mechanisms used by *A. alternata* to infect plants and inhibit other pathogens competitors are based on the synergism between enzymatic activities and the secretion of the non-host toxin. In fact, compounds **2** and **4** have been shown to play a key role in the mechanism used by *A. alternata* to decrease *B. cinerea* growth. Other modes of action, such as nutrient competition, reduction of competitor spore germination percentage, and induction of plant systemic resistance [54], could be involved in the *B. cinerea* inhibition and will be the focus of future studies.

## Figures and Tables

**Figure 1 jof-09-00326-f001:**
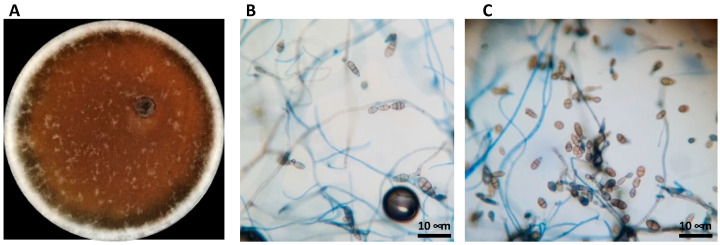
Morphological identification of *Alternaria alternata.* (**A**) *A. alternata* grown in PDA broth for 7 days at 28 °C. (**B**,**C**) Observations under phase-contrast light microscope magnification 60×.

**Figure 2 jof-09-00326-f002:**
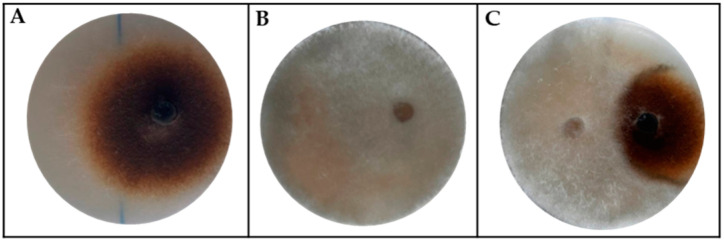
Competitive exclusion between pathogen fungi that infect the same host plant. (**A**) *Alternaria alternata*; (**B**) *Botrytis cinerea*; (**C**) Inhibition plate.

**Figure 3 jof-09-00326-f003:**
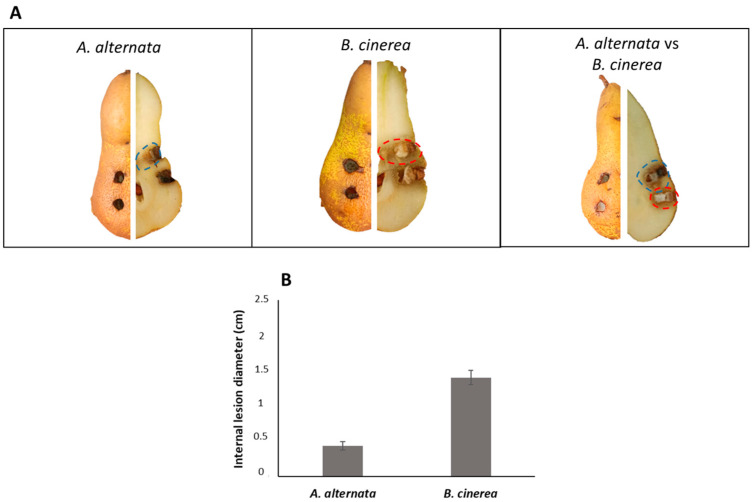
Pathogenicity assays on host Abate Fetel pear fruits. (**A**) Pathogenicity test on pear fruits; (**B**) severity (cm) of internal lesions on pear fruits.

**Figure 4 jof-09-00326-f004:**
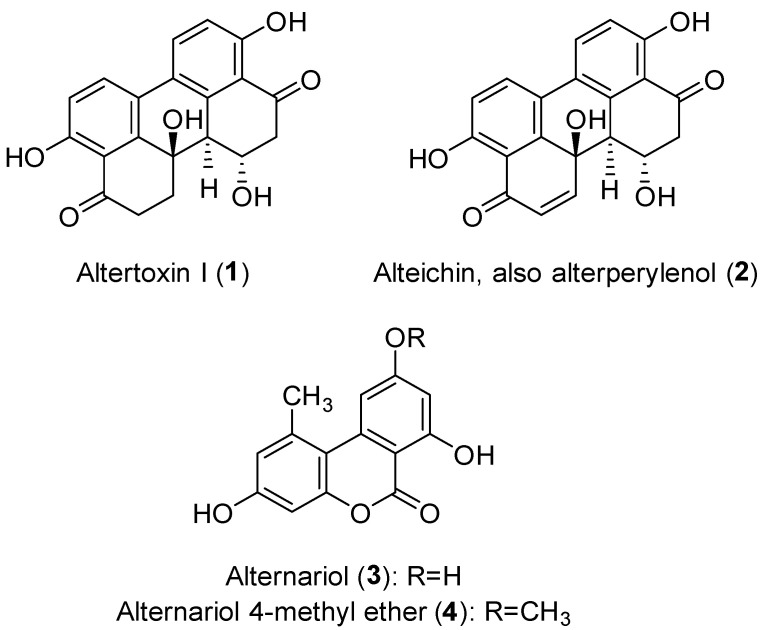
Chemical structure of the isolated metabolites.

**Figure 5 jof-09-00326-f005:**
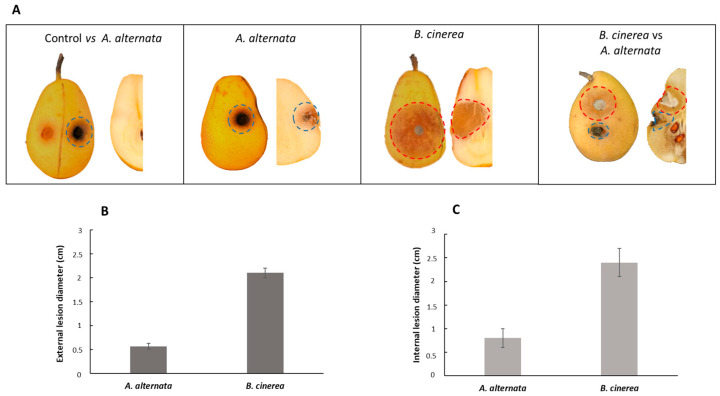
Pathogenicity assays on non-host Decana del Comizio pear fruits. (**A**) Pathogenicity test on pear fruits; (**B**) detection of external lesions on pear fruits; (**C**) detection of internal lesions on pear fruits.

**Figure 6 jof-09-00326-f006:**
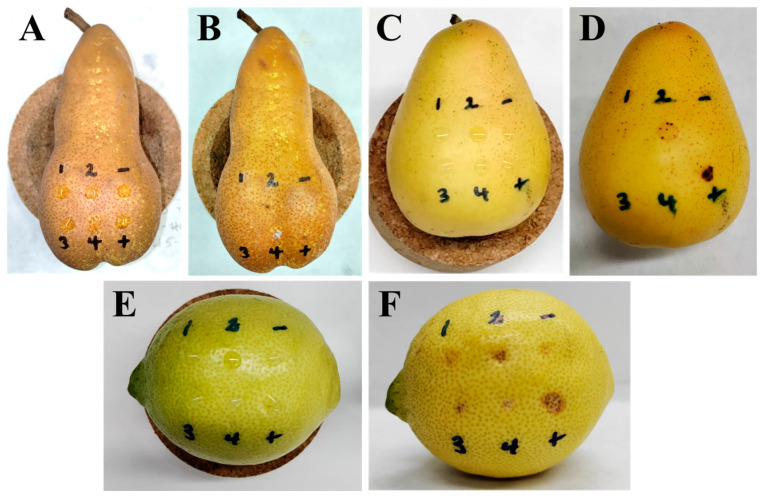
Necrotic effects produced by compounds **1**–**4** (tested at 5 × 10^−3^ M), the positive control *epi*-pyriculol (+) (tested at 5 × 10^−3^ M), and the negative control (-) (4% DMSO in distilled water), on Abate Fetel pears after 0 (**A**) and 3 (**B**) days; on Decana del Comizio pears after 0 (**C**) and 3 (**D**) days; and on lemon fruits after 0 (**E**) and 3 days (**F**).

**Figure 7 jof-09-00326-f007:**
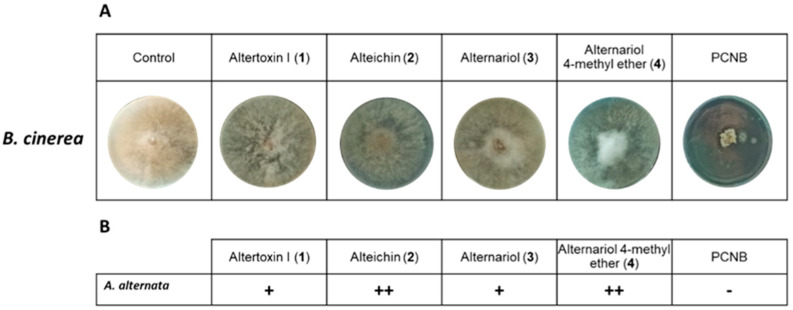
In vitro antagonistic effect of compounds isolated from *A. alternata*. (**A**): representative photographs of antifungal activity against *B. cinerea* of compounds **1**–**4** isolated from *A. alternata* compared with the positive control pentachloronitrobenzene (PCNB) at a final concentration of 5 × 10^−3^ M. (**B**): evaluation of inhibition growth: +: low inhibition; ++: medium inhibition; -: no growth.

**Table 1 jof-09-00326-t001:** List of the phytopathogenic fungi used in this study.

Species	Strain	Provenience
*Alternaria alternata*	RS	Italy
*Botrytis cinerea*	B05.10	Italy

**Table 2 jof-09-00326-t002:** Enzymatic activities of *Alternaria alternata*: Undetectable enzyme (-), halo 5 mm (++), and halo > 5 mm (+++).

Strain	Protease	Amylase	Xylanase	Cellulase
*Alternaria alternata*	-	+++	+++	++

**Table 3 jof-09-00326-t003:** Phytotoxic activity of compounds **1**–**4**, and positive (*epi*-pyriculol) and negative controls, on pear and lemon samples after 3 days at a concentration of 5 × 10^−3^ M.

Compound	Activity ^1^ on Abate Fetel Pears	Activity ^1^ on Decana del Comizio Pears	Activity ^1^ on Lemon Fruits
Altertoxin I	0	1	2
Alteichin	1	2	3
Alternariol	0	0	3
Alternariol 4-methyl ether	0	0	1
Positive control	1	3	3
Negative control	0	0	0

^1^ Intensity of necrosis is reported as: 3, severe necrosis; 2, intermediate necrosis; 1, slight necrosis; 0, no necrosis.

## Data Availability

Not applicable.

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
