# Peer review of "Alternaria alternata Isolated from Infected Pears (Pyrus communis) in Italy Produces Non-Host Toxins and Hydrolytic Enzymes as Infection Mechanisms and Exhibits Competitive Exclusion against Botrytis cinerea in Co-Infected Host Fruits"

_jof, 2023, doi:10.3390/jof9030326_

Round 1

Reviewer 1 Report

Line 44. you list only plants of agronomic interest, no one is ornamental.  Please rephrase like ..plant species including agronomically important ones like potato etc.

Line 52-53. Give at first mentioning the name of the toxin with abbreviations.  In the abstract you name them, but the abbreviations were not given. I understand, number of words, but here it would have its place.  Does this mean that the different toxin producing lines  are specialized pathogens for the given plant? This is an important point as you story only one single isolate that needs a more exact characterization.

Line 88. Here you should give the name of the toxins this isolate produces. This is important as your results will be valid only for the A alternata lineage and not all different member of the species that seems a diffuse mixture of a number of genotypes. Is this true or a hypothesis that need support? In this respect we need to know the cross reaction with other A alternata lineages infecting other plant species.  It is not a problem when we do not know it, but this should be communicated.

Line 119. How many Petri dishes were used  for measuring enzymatic activity? Please give the number. It is important to demonstrate repeatability of the tests. In this case also a variance could have an importance.

Line 130.  I realized that you tested during this study only one single isolate from pear. You had to have a standard isolate  to use by A. alternata to demonstrate similarity or agreements between the two fungal strains. This would have been highly important that you described rathe large difference between lines of A. alternata producing special toxins. To compere with B. cinerea does not seem me to be well founded.

Line 148. How many fruits were inoculated for a treatment? Was the inoculation made wit A. alternata and B cinerea? I guess, this is the case. We need a correct experimental design that excludes misunderstanding.  

Line 152. Where the samples for analyses came from? This is not clear. Do you have replicates to see the stability of the reaction?

Line 166. Does this mean that you analyzed only one flask of culture filtrate? How can you state a repeatability of your results?   

Line 196. What is DMSO? The experimental design should be clear. how many fruits you had in a replicate?

I have the impression that the authors overlooked several methodical rules to prove that repeatability of the results is essential. The basic idea is good. However, they had only one isolate for testing, even they stated that specialized lines for the different toxins exist.  By this way we cannot compare the data of this isolate to a generally accepted control strain. It is not clear, why Botrytis cinerea was co-tested. Did you have the impression that they could be antagonists? Even so, a data basis for a statistical analysis what is significant and what is not should have been presented.

At the present readiness of the paper, I do not find any reason to accept the data proven, therefore I cannot suggest its acceptance. I have the feeling that JoF needs a better planned and better performed experimental series with solid scientific proofs.  

Author Response

  I would like to thank the Reviewer for the useful suggestions

 All comments and suggestions have been considered and we feel they helped in improving the quality of the manuscript.

specifically:

Point-by-point reply

Line 44. you list only plants of agronomic interest, no one is ornamental.  Please rephrase like ..plant species including agronomically important ones like potato etc.

REPLY: Done.

Line 52-53. Give at first mentioning the name of the toxin with abbreviations.  In the abstract you name them, but the abbreviations were not given. I understand, number of words, but here it would have its place.

REPLY: We thank the Reviewer for the comment. However, “AK”, “AAL” and “AF” are not abbreviations but the name of toxins. The names of the toxins which we identified are altertoxin I (1), alteichin (2), alternariol (3), and alternariol 4-methyl ether (4).

 Does this mean that the different toxin producing lines  are specialized pathogens for the given plant? This is an important point as you story only one single isolate that needs a more exact characterization.

REPLY: Seven variants, designated as pathotypes, of the fungus Alternaria alternata have been identified, able to produce host-specific toxins  (Kohmoto et al. 1995). We agree with the reviewer on the interest of A. alternata pathotypes as a fascinating case for studying intraspecific variation in plant pathogenic fungi. Nevertheless, this manuscript aims not to identify the pathotype of the A. alternata field isolate but to investigate the unspecific toxins and their effect on the fungal competitors.

Line 88. Here you should give the name of the toxins this isolate produces. This is important as your results will be valid only for the A alternata lineage and not all different member of the species that seems a diffuse mixture of a number of genotypes. Is this true or a hypothesis that need support? In this respect we need to know the cross reaction with other A alternata lineages infecting other plant species.  It is not a problem when we do not know it, but this should be communicated.

REPLY: We apologize but don't understand where to insert information about toxins. In line 88, we describe the infected plant from where the A. alternaria strain was collected. Then the fungal field isolate was identified as A.alternata by morphologic characterization (paragraph 2.2) and sequencing of 18S (paragraph 2.3). Moreover, the secreted toxins, purified and identified, as reported in section 2.8, confirmed our identification. In addition, in section 3.5 of the discussion of results, the information on the toxins studied in our study is contextualized, providing adequate references for further expansion. However, as mentioned before, we focused our attention on the unspecific infection strategy of the isolated strain and not on the pathotype of the A. alternata.

Line 119. How many Petri dishes were used  for measuring enzymatic activity? Please give the number. It is important to demonstrate the repeatability of the tests. In this case also a variance could have an importance.

REPLY: The Reviewer is right. Each experiment was triplicate conducted. We have added this information in the material and methods section, paragraph 2.4.

Line 130.  I realized that you tested during this study only one single isolate from pear. You had to have a standard isolate  to use by A. alternata to demonstrate similarity or agreements between the two fungal strains. This would have been highly important that you described rathe large difference between lines of A. alternata producing special toxins. To compere with B. cinerea does not seem me to be well founded.

REPLY: Unfortunately, a standard or an ATCC strain of A. alternata is not available, making any comparison difficult. Moreover, it is known that microorganisms' pathogenicity activity is influenced by the host plants and the geographical areas in which they live. So we decided to compare the A. alternata field isolate with  B. cinerea, since competitors for the same plant, the pear, and sourced from the same region with respect to another Alternaria spp.

Line 148. How many fruits were inoculated for a treatment?

REPLY:  As reported in section 2.6: "Each treatment was conducted with three replications and repeated three times.”

Was the inoculation made wit A. alternata and B cinerea? I guess, this is the case. We need a correct experimental design that excludes misunderstanding.  

REPLY:  The reviewer is right. The information has been added to the text (lines 174-177).

Line 152. Where the samples for analyses came from? This is not clear. Do you have replicates to see the stability of the reaction?

REPLY: We apologize but don't understand “the stability of the reaction”. No reactions were performed in this study, but the samples analyzed by NMR and MS were isolated from the culture filtrate of the fungus A. alternata, as described in section 2.8.

Line 166. Does this mean that you analyzed only one flask of culture filtrate? How can you state a repeatability of your results?   

REPLY: The supernatants obtained from three different fungal growth were analyzed, and we obtained the same compounds pattern.

Line 196. What is DMSO? The experimental design should be clear. how many fruits you had in a replicate?

REPLY: we thank the reviewer for the useful corrections. The extension of the DMSO name (line 225) and the number of fruits used in each replicate  (line 227) have been added in the material and method section.

I have the impression that the authors overlooked several methodical rules to prove that repeatability of the results is essential. The basic idea is good. However, they had only one isolate for testing, even they stated that specialized lines for the different toxins exist.  By this way we cannot compare the data of this isolate to a generally accepted control strain. It is not clear, why Botrytis cinerea was co-tested. Did you have the impression that they could be antagonists? Even so, a data basis for a statistical analysis what is significant and what is not should have been presented.

At the present readiness of the paper, I do not find any reason to accept the data proven, therefore I cannot suggest its acceptance. I have the feeling that JoF needs a better planned and better performed experimental series with solid scientific proofs.  

REPLY: We agree on the lack of information about the use of B. cinerea, and we thank the reviewer for the useful suggestion. We have reorganized the introduction, adding important information necessary to clarify our objectives. Regarding the "methodical rules" contestation, each experiment was performed in triplicate, and the statistical analysis is reported in the material and method section. Moreover, the absence of an ATTC strain of A. alternata isolated from Pyrus communis fruit makes a comparison between our isolated and control strain impossible.

Reviewer 2 Report

The manuscript reports a study of enzymatic activity to produce non-specific toxins metabolites and degrade cell walls, and co-infection together with B. cinerea by  A. alternata isolated from infected pears.

1     The "lack of information" motivation for the study given on line 72 is supported by reference 11 from 2009. This is somewhat dated and should be replaced by
H Wang et al., Recent Advances in Alternaria Phytotoxins: A Review of Their
Occurrence, Structure, Bioactivity, and Biosynthesis, Journal of Fungi 8 (2022) 168, which in section 3 supports the claim of non-host selective toxins receiving less attention.

2     A historic perspective could be added in the introduction based on information in M Tanahashi et al., Black spot disease of European pear [Pyrus communis], a new disease caused by Alternaria alternata (Fries: Fries) Keissler, Annals of the Phytopathological Society of Japan 70 (2004) 168-175. Included could be information for non-specialists such as why the disease was discovered in Japan and apparently only originally infected Le Lectier and General Leclerc. Also if there is any significance of the black, brown or gray designations of spots for the specific species involved in the study.

3     The statement on line 288 about the difference in competition between host and in vitro would seem to need some follow-up discussion, such as if it is a general phenomenon or specific to the pathogens under study.

Also of importance with respect to postharvest infection is maturity stage of the fruit. This parameter is not addressed in any standard experimental protocol. The procedure in reference 22 which was followed in the manuscript described obtaining healthy fruit from a market. In M C Lutz et al., Antifungal effects of low environmental risk compounds on development of pear postharvest diseases: Orchard and postharvest applications, Scientia Horticulturae 295 (2022) 110862, the experiments were performed on healthy Pyrus communis L. after 30 days in cold storage.

The question of if fruit maturity played a role in the B. cinerea and A. alternata infection mechanisms probably can't be answered. But a comparison between the two studies done a half hemisphere apart shows similar results. The controls with no added antifungal compound in Table 4 of the Lutz article show fruit lesion severity of 6 mm for Alternaria alternata and 80 mm for Botrytis cinerea, a much larger difference ratio than the 4 shown in Figure 6B.

4     While it is unlikely the conclusions drawn from the study can be applied beyond the specific species involved, a short discussion of how broad an application of results may be possible would be of interest.

The introduction in A Ortuno et al., Correlation of ethylene synthesis in Citrus fruits and their susceptibility to Alternaria alternata pv. citri, Physiological and Molecular Plant Pathology 72 (2008) 162-166, discussed that susceptibility of cultivars to A alternata is high for  Citrus reticulata, medium for Citrus clementina Hort. ex Tan. and low for Citrus limon (L.) Burm.

There are thus a number of points which could possibly be addressed concerning how the results shown in Table 2 for alteichin could be more broadly applicable, among them: If susceptibility of fruit gives the same susceptibility of cultivars, if the result would be the same if the fruit were Citrus limon (L.) Burm., if the lower concentration of 2.5x10-3 M would give measurable activity for highly susceptible Citrus reticulata, and if alteichin from another Alternata source such as pv. citri would have the same level 3 necrosis effect on the Citrus limon studied?    

5     Along with maturity, whether the fruit is wounded or unwounded seem to be optional parameters. That the pears in table 2 were unwounded and the lemons punctured had a significant effect on results.

Related to this is that cellulase activity was determined but not cutinase. Was this because of the time factor, that there will always be wounds on fruits and to enter through the cuticle would take a prohibitively long time?

Author Response

I want to thank the Reviewer for the useful suggestions. All comments and suggestions have been considered, and we feel they helped in improving the quality of the manuscript.

specifically:

Point-by-point reply

1     The "lack of information" motivation for the study given on line 72 is supported by reference 11 from 2009. This is somewhat dated and should be replaced by

H Wang et al., Recent Advances in Alternaria Phytotoxins: A Review of Their

Occurrence, Structure, Bioactivity, and Biosynthesis, Journal of Fungi 8 (2022) 168, which in section 3 supports the claim of non-host selective toxins receiving less attention.

REPLY: we thank the reviewer for the useful advice. The reference has been replaced.

2     A historic perspective could be added in the introduction based on information in M Tanahashi et al., Black spot disease of European pear [Pyrus communis], a new disease caused by Alternaria alternata (Fries: Fries) Keissler, Annals of the Phytopathological Society of Japan 70 (2004) 168-175. Included could be information for non-specialists such as why the disease was discovered in Japan and apparently only originally infected Le Lectier and General Leclerc. Also if there is any significance of the black, brown or gray designations of spots for the specific species involved in the study.

REPLY: We thank the reviewer for the useful suggestion. This information is added in the introduction (lines 47-50). We prefer not to add a paragraph on "the significance of the black, brown or gray designations of spots for the specific species " and not to stretch the introduction section too much.

3     The statement on line 288 about the difference in competition between host and in vitro would seem to need some follow-up discussion, such as if it is a general phenomenon or specific to the pathogens under study.

REPLY: the pathogens' Interactions in Co-infected Plants have been discussed more deeply in the text.

Also of importance with respect to postharvest infection is maturity stage of the fruit. This parameter is not addressed in any standard experimental protocol. The procedure in reference 22 which was followed in the manuscript described obtaining healthy fruit from a market. In M C Lutz et al., Antifungal effects of low environmental risk compounds on development of pear postharvest diseases: Orchard and postharvest applications, Scientia Horticulturae 295 (2022) 110862, the experiments were performed on healthy Pyrus communis L. after 30 days in cold storage.

REPLY: The Reviewer is right. We have added the missing information about the pears in the material and methods section. We have used the methodology described in reference 22. So, the fruits were obtained from the fruit seller in Naples (Italy), and the pathogenicity test was performed on healthy fruit after 2 days in cold storage.

The question of if fruit maturity played a role in the B. cinerea and A. alternata infection mechanisms probably can't be answered.

But a comparison between the two studies done a half hemisphere apart shows similar results. The controls with no added antifungal compound in Table 4 of the Lutz article show fruit lesion severity of 6 mm for Alternaria alternata and 80 mm for Botrytis cinerea, a much larger difference ratio than the 4 shown in Figure 6B.

REPLY: We agree with the reviewer. It's very difficult to respond if the fruit maturity influences the infection mechanisms because a lot of variabilities are implicated, such as inoculum pressure, timing to application, the virulence of the strains used, and the size of the artificial wounds. They are all factors implying critical differences with what occurs naturally. However, we thank the reviewer for the useful suggestion, and we have added the information presented in the article Lutz et al. in the introduction and results section.

4     While it is unlikely the conclusions drawn from the study can be applied beyond the specific species involved, a short discussion of how broad an application of results may be possible would be of interest.

The introduction in A Ortuno et al., Correlation of ethylene synthesis in Citrus fruits and their susceptibility to Alternaria alternata pv. citri, Physiological and Molecular Plant Pathology 72 (2008) 162-166, discussed that susceptibility of cultivars to A alternata is high for  Citrus reticulata, medium for Citrus clementina Hort. ex Tan. and low for Citrus limon (L.) Burm.

There are thus a number of points which could possibly be addressed concerning how the results shown in Table 2 for alteichin could be more broadly applicable, among them: If susceptibility of fruit gives the same susceptibility of cultivars, if the result would be the same if the fruit were Citrus limon (L.) Burm., if the lower concentration of 2.5x10-3 M would give measurable activity for highly susceptible Citrus reticulata, and if alteichin from another Alternata source such as pv. citri would have the same level 3 necrosis effect on the Citrus limon studied?   

REPLY: We thank the reviewer for the comment. This point has been clarified in the text. Regarding the last question, compounds with the same chemical structure and stereochemistry isolated from different sources or obtained by synthesis show the same activity on the same target and at the same concentration. Moreover, a potential application of the identified toxins has been inserted in the text (lines 499-503)

5     Along with maturity, whether the fruit is wounded or unwounded seem to be optional parameters. That the pears in table 2 were unwounded and the lemons punctured had a significant effect on results.

Related to this is that cellulase activity was determined but not cutinase. Was this because of the time factor, that there will always be wounds on fruits and to enter through the cuticle would take a prohibitively long time?

REPLY: We thank the reviewer for the comment. In previous studies, we experimentally observed necrosis on wounded pears with negative control but not with unwounded one. Thus, unwounded pears are needed to obtain reliable results.

Investigation of the hydrolysis activity and HST toxins production of the A. alternata RS will be the topic of future work.

Reviewer 3 Report

I found that the information contained in the manuscript seems to be disordered.

The selected title does not truly represent the contents mentioned in the manuscript. The central place in Results has data on competitive exclusion between pathogen fungi that infect the same host plant, Alternaria alternata and Botrytis cinerea, and the title must reflect it.

Keywords could be revised (+ Botrytis cinerea etc)

 Abstract could be more concise as well as future implications can be provided at the end of the abstract section (Postulate the main problem. Is it inhibition of Botrytis by Alternaria toxins? Give a concise aim and conclusion, “These results suggest a different mechanism of competition between the two pathogens on host pears and in vitro.” is not suited for this journal).

 In the introduction section, I found the text lacking any information on the problem of multiinfection of fruits with some pathogens, about Botrytis pathogenesis etc. A lot of literature data didn’t have references. Some of them (For example, “All the HSTs have different modes of action, causing biochemical and genetic modifications in the host.”) need verifying and must be considered in more detail. Then authors said about Host and non-host specific toxins they should give information on Z-scheme of plant -pathogen interaction.

The aim “In this context, our work reports a study focused on NSTs and CWDEs used by an A. alternata strain isolated from infected pears in Italy” does not truly represent the content of the results obtained (pay attention to Botrytis).

 Results: Figure 2 is table 2, figure 3 is figure 2 etc.

 Fig. 2 Competitive exclusion between pathogen fungi that infect the same host plant: I see that Botrytis grow quicker than Alternaria and occupy the whole place. Provided photo of inhibition plate then between colonies was an empty place and Alternaria colony was smaller than Botrytis.

Fig 4, 6 Mark Alternaria and Botrytis necrosis on one pear. Did you analyze the depth of lesions in tissues?

 Tables in figures do not give important information and should be deleted.

 Fig 7. Necrotic effects produced by compounds 1-4 on Abate Fetel pears after 0 (a) and 3 (b) days; onDecana del Comizio pears after 0 (c) and 3 (d) days; and on lemon fruits after 0 (e) and 3 days (f). Are you sure that non-host toxins led to the development of necrosis, not hyper-sensitive reactions?

Conclusion: Please remove needless statements. Your thesis on B. cinerea inhibits A. alternata growth in vitro is based on questionable data. You can delete this data and plate tests and discuss the basic principles of fungi interaction in plants and finish the article properly (not “These toxins slow down the extension of the fungal hyphae which is beneficial for the uptake of nutrients.”, may be future perspectives).

 Double check that all references are cited within the text, and that all citations within the text have a corresponding reference.

Author Response

I want to thank the Reviewer for the useful suggestions. All comments and suggestions have been considered and we feel they helped in improving the quality of the manuscript.

specifically:

Point-by-point reply

I found that the information contained in the manuscript seems to be disordered.

REPLY: We reviewed the article based on your suggestions.

The selected title does not truly represent the contents mentioned in the manuscript. The central place in Results has data on competitive exclusion between pathogen fungi that infect the same host plant, Alternaria alternata and Botrytis cinerea, and the title must reflect it.

REPLY: We thank the reviewer for the useful observations; the title has been modified according to the suggestion as: “Alternaria alternata Isolated from Infected Pears (Pyrus communis) in Italy produces Non-Host Toxins and hydrolytic enzymes as infection mechanism and exhibits competitive exclusion against Botrytis cinerea in co-infected host fruits.”

Keywords could be revised (+ Botrytis cinerea etc)

REPLY: Done.

Abstract could be more concise as well as future implications can be provided at the end of the abstract section (Postulate the main problem. Is it inhibition of Botrytis by Alternaria toxins? Give a concise aim and conclusion, “These results suggest a different mechanism of competition between the two pathogens on host pears and in vitro.” is not suited for this journal).

REPLY: We thank the reviewer for the useful observations. The abstract has been summarized with a better understanding of the aim and results. The toxins secreted by A. alternata are involved in the competitive exclusion, as reported in Fig. 8, but it's not possible to attribute this activity only to them.

 In the introduction section, I found the text lacking any information on the problem of multiinfection of fruits with some pathogens, about Botrytis pathogenesis etc. A lot of literature data didn’t have references. Some of them (For example, “All the HSTs have different modes of action, causing biochemical and genetic modifications in the host.”) need verifying and must be considered in more detail. Then authors said about Host and non-host specific toxins they should give information on Z-scheme of plant -pathogen interaction.

REPLY: The Reviewer is right. The pathogens' Interactions in Co-infected Plants have been discussed more deeply in the text, and Botrytis cinerea information (lines 84-94) and the missed references have been added.

The aim “In this context, our work reports a study focused on NSTs and CWDEs used by an A. alternata strain isolated from infected pears in Italy” does not truly represent the content of the results obtained (pay attention to Botrytis).

REPLY: The Reviewer is right. We have rewritten the aims of our work, and we hope that now they are clearer

Results: Figure 2 is table 2, figure 3 is figure 2 etc.

REPLY: Done

 Fig. 2 Competitive exclusion between pathogen fungi that infect the same host plant: I see that Botrytis grow quicker than Alternaria and occupy the whole place. Provided photo of inhibition plate then between colonies was an empty place and Alternaria colony was smaller than Botrytis.

REPLY: The reviewer is right. A. alternata grow a little slower than B. cinerea. We stopped the experiment when the fungi expanded on the entire co-inoculation plate according to the material and method section. However, if A. alternata had been able to inhibit B. cinerea in the co-inoculation plates, the results would have been like on the pear fruits assays. On the contrary, we have obtained a 44% growth inhibition of A. alternata concerning the fungus grown alone. The result confirms our hypothesis. A. alternata in PDA broth is not able to perform the antifungal activity.

Fig 4, 6 Mark Alternaria and Botrytis necrosis on one pear.

REPLY: DONE

Did you analyze the depth of lesions in tissues? Tables in figures do not give important information and should be deleted.

REPLY: Yes, we did. We have measured only the internal lesion for the results shown in Figure 4. For those shown in Figure 6, we have added the results on the internal lesion instead of the table you advised us to delete. The table has also been deleted in Figure 4.

Fig 7. Necrotic effects produced by compounds 1-4 on Abate Fetel pears after 0 (a) and 3 (b) days; onDecana del Comizio pears after 0 (c) and 3 (d) days; and on lemon fruits after 0 (e) and 3 days (f). Are you sure that non-host toxins led to the development of necrosis, not hyper-sensitive reactions?

REPLY: We have identified the activity of toxins as necrosis on the fruits based on morphological observation and literature studies that attribute this activity to NST toxins secreted by A. alternata (https://doi.org/10.3390/jof8020168).

Conclusion: Please remove needless statements. Your thesis on B. cinerea inhibits A. alternata growth in vitro is based on questionable data. You can delete this data and plate tests and discuss the basic principles of fungi interaction in plants and finish the article properly (not “These toxins slow down the extension of the fungal hyphae which is beneficial for the uptake of nutrients.”, may be future perspectives).

REPLY: The conclusion section has been improved. We have not deleted the data about the B. cinerea- A. alternaria interactions. To our knowledge, the interaction tests between these two pathogens have not been performed yet. Even if our results are preliminary, we believe they are interesting and will be the beginning of future studies. We hope that the revised conclusion properly addresses the issues raised by the reviewer.

 Double check that all references are cited within the text, and that all citations within the text have a corresponding reference.

REPLY: Done

Reviewer 4 Report

A brief summary

The phytopathogenic fungus isolated from infected pear fruits was identified as Alternaria alternate. Four toxic compounds (altertoxin I, alteichin, alternariol and alternariol 4-methyl ether) have been isolated from the cultivation medium of this fungus with different level of activity. Only alteichin caused necrotic lesions on host pears, while altertoxin I and alteichin showed necrotic activity on non-host pears. All four compounds caused necrotic lesions on other non-host fruits (lemon, Citrus limon). It was shown that two pathogenic fungi (A. alternate and Botrytis cinerea) demonstrate different mechanisms of competition between themselves on the pear host and in vitro.

Broad comments 

Good, well-written work, application of microbiological, biochemical, physico-chemical methods, determination of competitive inhibition of two fungi under different conditions. It is of undoubted interest for specialists in pathogenic fungi of fruit crops.

Specific comments 

Line 25: - “...from its PDB liquid cultures…”   change to   “…from its potato dextrose broth liquid cultures…”

Line 163: - “PDB (Difco, USA)”   change to   potato dextrose broth (PDB) (Difco, USA)”

Line 468:   https://www.mdpi.com/xxx/s1  - this link not working. Therefore, I could not get acquainted with additional materials.

Author Response

  I would like to thank the Reviewer for the useful suggestions. All comments and suggestions have been considered and we feel they helped in improving the quality of the manuscript.

specifically:

Point-by-point reply

Good, well-written work, application of microbiological, biochemical, physico-chemical methods, determination of competitive inhibition of two fungi under different conditions. It is of undoubted interest for specialists in pathogenic fungi of fruit crops.

REPLY: We thank the Reviewer for the positive comment

Line 25: - “...from its PDB liquid cultures…”   change to   “…from its potato dextrose broth liquid cultures…”

REPLY: Done

Line 163: - “PDB (Difco, USA)”   change to   “potato dextrose broth (PDB) (Difco, USA)”

REPLY: Done

Line 468:   https://www.mdpi.com/xxx/s1  - this link not working. Therefore, I could not get acquainted with additional materials.

REPLY: This link regards the supplementary materials assigned if the article is published. We have uploaded the supplementary materials file together with the article for revision. Therefore, it is possible to download the file in the "Supplementary File" section.

Round 2

Reviewer 1 Report

The line 102 was not corrected, the toxin producing capacity of the B. cinerea?  or A. alternata?  strain was not given. Instead of „fungal pathogen” is B cinerea better, I see in line 106 that three strains were given to culture collection, and not one, as indicated.  This should be clarified.

Line 142. I would suggest giving the experimental design. The question is that you made the enzyme measurement from each Petri dish, or you pooled the three Petri dishes and made one measurement from them. This is important.

Line 168. Int he previous opinion I mentioned that no control isolate was tested for the two pathogens, so the possible difference between the new selected strain and the representative  species isolate could not be compared,  This problem remained. In the line 168 the control as treatment has not been mentioned.  Here also the question is, whether the replicates were  evaluated for each replicate separately or pooled. One thing should be clarified, what does it men a replicate, one or more fruits?

Line 184. I see, the new sentence was inserted, It would be better to place this sentence  as first sentence following the 2.8 title line.  It should be mentioned that it comes from the procedure  described in 2.7.

Line 187. It seems that for toxin analysis only one 1 2 L unit culture filtrate was used. For this remark I did not receive a response.  How can you prove the repeatability of your experimental results by this way?

I think that the general outcome of the opinion in spite of the many corrections that improved the paper, but the basic problems could not be solved. For this reason, I cannot support the publication of the paper. I think that the problem is vivid, the idea is good, the tests should be extended and repeated. I would like to see this new paper that will be suitable for acceptance. I know from own experience that a rejection of a paper does not cause happiness. However, we all have the same interest, to publish the best possible paper.

Author Response

The line 102 was not corrected, the toxin producing capacity of the B. cinerea?  or A. alternata?  strain was not given. Instead of „fungal pathogen” is B cinerea better, I see in line 106 that three strains were given to culture collection, and not one, as indicated.  This should be clarified.

REPLY: As already reported in our “point-to-point reply” to the Reviewer about the review of line 88 (now 120), in paragraph 2.1 of Material and Methods, we described the isolation of a fungal phytopathogen from an infected plant of Pyrus communis. This fungal field isolate was identified as A.alternata RS as reported in paragraph 2.3.

we replaced  „fungal pathogen” with Alternata alternaria RS. Moreover, line 106 reports the Agriges s.r.l coordinate, where the strain was isolated and stored. We have deleted this information to avoid misunderstanding.

Line 142. I would suggest giving the experimental design. The question is that you made the enzyme measurement from each Petri dish, or you pooled the three Petri dishes and made one measurement from them. This is important.

REPLY: We thank the reviewer for the useful suggestion. We have modified the text to clarify the protocol used (lines 143-144). In particular, each test was triplicate conducted and repeated three times, and the values reported in table 2 are the mean of the halos’ values obtained.

Line 168. Int he previous opinion I mentioned that no control isolate was tested for the two pathogens, so the possible difference between the new selected strain and the representative  species isolate could not be compared,  This problem remained.

REPLY: To the best of our knowledge, a fungal strain of A. alternata isolated from Abate Fetel pear tree  is not available. Moreover, as previously reported, a standard or an ATCC strain of this fungal specie is not present. So we were not able to compare A. alternata RS field strain withthe  one of control. We thought it would be interesting to study field strains able to infect the same plant.

In the line 168 the control as treatment has not been mentioned.  

REPLY: The control used in this experiment is a PDA plug, used to grow the fungi  according to the reference used (Aung et al. (2020).

Here also the question is, whether the replicates were  evaluated for each replicate separately or pooled. One thing should be clarified, what does it men a replicate, one or more fruits?

REPLY: As reported in line 165 (in the new version, line 170), “Each treatment was conducted with three replications and repeated three times.” Figures 3 and 5 report the means of decay lesion diameters measured on nine fruits for each treatment. A sentence has been added to clarify this point (line 71).

Line 184. I see, the new sentence was inserted, It would be better to place this sentence  as first sentence following the 2.8 title line. It should be mentioned that it comes from the procedure  described in 2.7. 

REPLY:  We have modified the text following the reviewer's suggestions (lines 200-203). Please,  see  the next point 

Line 187. It seems that for toxin analysis only one 1 2 L unit culture filtrate was used. For this remark I did not receive a response.  How can you prove the repeatability of your experimental results by this way?

REPLY: The reviewer is right. We apologize for not being clear in our previous revision. Other two A. alternaria growths in 1 L of PD broth have been performed, and the chromatographic pattern of the organic extracts showed the same metabolites (toxins 1-4). This last sentence has been added to the text (lines 207-209).

I think that the general outcome of the opinion in spite of the many corrections that improved the paper, but the basic problems could not be solved. For this reason, I cannot support the publication of the paper. I think that the problem is vivid, the idea is good, the tests should be extended and repeated. I would like to see this new paper that will be suitable for acceptance. I know from own experience that a rejection of a paper does not cause happiness. However, we all have the same interest, to publish the best possible paper.

REPLY: We thank the revisor for the suggestions and for empathy if our work is rejected. We have modified the text, clarifying all the raised points. We hope that the reviewer can appreciate our improvements.